# Chemical Composition of Leaves, Stem, and Roots of *Peperomia pellucida* (L.) Kunth

**DOI:** 10.3390/molecules27061847

**Published:** 2022-03-11

**Authors:** Paulo Wender P. Gomes, Hugo Barretto, José Diogo E. Reis, Abraão Muribeca, Alice Veloso, Carlos Albuquerque, Andrew Teixeira, Wandson Braamcamp, Sônia Pamplona, Consuelo Silva, Milton Silva

**Affiliations:** 1Collaborative Mass Spectrometry Innovation Center, Skaggs School of Pharmacy and Pharmaceutical Sciences, University of California San Diego, La Jolla, CA 92093, USA; wendergomes@ufpa.br; 2Laboratory of Liquid Chromatography (Labcrol), Institute of Exact and Natural Sciences, Federal Universityof Pará, Belém 66075-110, Brazil; hugorbarretto@gmail.com (H.B.); reisdiogo190@gmail.com (J.D.E.R.); abraao_muribeca@hotmail.com (A.M.); alice19carvalhos@gmail.com (A.V.); carlos.albuquerque@icen.ufpa.br (C.A.); wbraamcamp@ufpa.br (W.B.); sgpamplona@yahoo.com.br (S.P.); yumikoyoshioka@yahoo.com.br (C.S.); 3Department of Natural Sciences, Faculty of Chemistry, Campus Barcarena, State University of Pará, R. Tomás Lourenço Fernandes, Barcarena 68445-000, Brazil; andrewmagno61@gmail.com; 4Faculty of Pharmaceutical Sciences, Federal University of Pará, Belém 66075-110, Brazil

**Keywords:** Piperaceae, LC-MS/MS, molecular network, machine learning, Sirius

## Abstract

*Peperomia pellucida* is a species known in the Amazon as “erva-de-jabuti” that has been used in several therapeutic applications based on folk medicine. Herein, we describe the classes, subclasses, and the main compounds of the leaves, stems, and roots from *P. pellucida* by ultra-high performance liquid chromatography coupled to high-resolution mass spectrometry associated with molecular networks, mirror plot on the GNPS library, and machine learning. These data show compounds that were annotated for the first time in the *Peperomia* genus, such as 2′,4′,5′-trihydroxybutyrophenonevelutin, dehydroretrofractamide C, and retrofractamide B.

## 1. Introduction

*Peperomia* is the second genus with the highest biological and chemical diversity in the family Piperaceae [1], with a Pantropical distribution it includes about 1500–1700 species [2], of which 90% occur in the Neotropical region [3]. In the list of the largest in the number of species in the world, the genus occupies the 19th position with the greatest diversity verified in the interior of tropical forests and expressive occurrence in the epiphytic habit [4]. In Brazil, the family is represented by 171 species distributed in the phytogeographic domains of the Amazon, Caatinga, Cerrado, and Atlantic Forest [5].

In traditional medicine, *Peperomia* species have been used to treat asthma, cough, ulcers, conjunctivitis, inflammation and high cholesterol and have functioned as diuretics, analgesics, and antibiotics [6]. Previous reports on species of the genus, revealed several biological activities, including antifungal [7], insecticidal [8], antiviral [9], anti-inflammatory [10], antileishmanial [11], and antiplasmodium [12]. Despite this, only 38 species were studied on the chemical profile, leading to the isolation of more than 200 compounds from different classes, highlighting the great chemodiversity of this genus [13].

*Peperomia pellucida* (L.) Kunth, popularly known in the Amazon as “erva-de-jabuti”, is a small herbaceous plant that occurs in countries in Asia, North, Central (Antilles), and South America [14]. In Brazil, it ranges from the Amazon to Paraná and is easily found in humid and shady environments [5]. In traditional medicine, it is widely used in South American and Asian countries due to its antihypertensive, antimicrobial, analgesic, anti-inflammatory, antipyretic, antidiabetic, and antioxidant properties [15,16]. These characteristics allow classifying this species as the most chemically studied within the genus *Peperomia* [17].

The main chemical profile reported to *P. pellucida* consists of alkaloids, flavonoids, sterols, tannins, reducing sugars, saponins, triterpenoids, carbohydrates, phenols, azulenes, carotenoids, depsides, and quinones [18,19]. They are associated with the analgesic, anti-inflammatory, antipyretic, bactericidal, and fungicidal potential of the species [20]. Other research also shows that *P. pellucida* may be a good natural source of antioxidants by suppressing oxidative stress in various metabolic diseases [21,22].

Despite all this, so far, no work has shown the chemical diversity in the different parts of this species and to support the discovery of new metabolites, recently the ultra-high performance liquid chromatography coupled to tandem mass spectrometry (UHPLC-MS/MS) associated with machine learning tools, for example, Molecular Network available in Global Natural Products Social Molecular Networking (GNPS), has been widely used to unravel the chemical diversity of natural products [23,24], providing useful information for structural characterization [25]. 

Considering the applicability of *P. pellucida* and the biological potential already reported in the literature, this research focuses on the metabolic diversity of different parts (leaves, stems, and roots) of this species, to contribute to the knowledge of the chemistry of this family and other species the genus. For this, the Feature-Based Molecular Networking (FBMN) tool was used to assist in the description of classes, subclasses, and some bioactive compounds, which were described for the first time in the leaves, stem, and root of *P. pellucida*, contributing to future studies, interpretations, and annotations of natural products with pharmacological potential [26,27].

## 2. Experimental Section

### 2.1. Plant Material

Leaves, stems, and roots of *P. pellucida* were collected at the Federal University of Pará 1°28′30″ S and 48°27′24″ W. All parts were sanitized with 70:30 ethanol (*w*/*w*, Minas Gerais, Brazil) to inhibit the proliferation of fungi and bacteria and subsequently washed with ultrapure water obtained in a Direct-Q 5 system (Millipore, Darmstadt, Germany). The resulting materials were dried in an oven with air circulation at 40 °C until constant weight (Quimis, Diadema, Brazil). An exsiccate was deposited in the herbarium of Embrapa Amazônia Oriental, registered by voucher IAN 197198. Lastly, the species was registered (A678D8C) with the National System for the Management of Genetic Heritage and Associated Traditional Knowledge (SISGEN, Brazil).

### 2.2. Extraction, Sample Preparation, and Isolation Process

Dried botanical materials were crushed in ball mills (Fritsch, Idar-Oberstein, Germany). After that, a total of 500 mg of each material was extracted with 10 mL of ethanol 99.8% (Minas Gerais, Brazil) using indirect ultrasound assisted extraction (Branson 2510, Danbury, CT, USA) [28] for 40 min based on the method available in the literature [29]. Subsequently, the volume was filtered and oven-dried (40 °C) until constant weight.

A pre-treatment was applied to remove interfering particles where 50 mg of the crude extract from the leaves, stems, or roots were solubilized in 5 mL of a mixture of water and acetonitrile (20:80). The resulting solution was homogenized in an ultrasonic bath (Branson **^®^**) for 1 min and applied to a solid-phase extraction (SPE) C18 cartridge (Phenomenex, Torrance, CA, USA) with 1 g of stationary phase. The samples were dried at 45 °C to constant weight. 

### 2.3. UHPLC-MS/MS Analysis

The LC-MS experiments were performed in a Xevo G2-S QTof a high-resolution mass spectrometer (Waters Corp., Milford, MA, USA) equipped with a Lockspray source. Leucine-enkephalin was used as reference for accurate mass measurements. MassLynx 4.1 software was used for system control and data acquisition. A total of 2 μL of extracts (2000 ppm) were injected and the chromatographic separation occurred in a BEH C18 column at 40 °C (Waters Corp; 50 mm; 2.1 mm; 1.7 μm particle size). Ultra-pure water (A) and acetonitrile (B) were used as the mobile phase (0.1% formic acid in both) to perform a method with a flow of 300 μL/min and a total time of 18 min in the following settings: 0–12 min, linear gradient from 5% to 95% B; 12–14, column cleaning; 14–15 min, a linear decrease from 95% to 5% B; and 15–18 min, held at 5% B for 3 min for equilibration of the column for the next experiment. The positive ionization (PI) was used to acquire the data in a mass range from *m*/*z* 50 to 1200. The settings of Data-Dependent Acquisition (DDA) experiments were used as follows: centroid format, number of ions selected 5 (Top5 experiment), the collision energy was set to 10, 20, 30, and 40. The scan time of 0.1 s, charge states of +1, tolerance window of ± 0.2 Da and peak extract window of 2 Da, tolerance of deisotope ± 3 Da, extraction tolerance of deisotope 6 Da. The source and desolvation temperatures were set to 150 °C and 300 °C. The cone and desolvation gas flow were set to 50 L/h and 800 L/h, respectively. The capillary and cone voltage were set at 3.0 kV and 40 V (in this order).

### 2.4. Compound Characterization

The files acquired in the Xevo G2-S QTof mass spectrometer to the ethanolic extracts were converted from raw into mzML format using MSConvert software (ProteoWizard, Palo Alto, CA, US) [30]. Posteriorly, the data were processed using MZmine software, version 2.53 [31]. The noise level for the detection of ions was set to 1.0 × 10^4^ (MS^1^) and 1.0 × 10^2^ (MS^2^). ADAP chromatogram was used to construct the total ion chromatogram (TIC) and a minimum group size of 3 and a minimum group intensity limit of 1.0 × 10^4^, a min highest intensity of 3.0 × 10^4^ and an *m*/*z* tolerance of 0.002 Da. The ADAP algorithm was used in the deconvolution process. The isotope detection was performed with an error of 10 ppm, RT tolerance of 0.2 min, and charge +1. For peak alignment, the tolerance of *m*/*z* 10 ppm was used, a score for *m*/*z* of 75 over 25 for RT with a tolerance of 0.2 min. The resulting list was filtered using a peak finder and the lines with no associated MS^2^ spectrum were removed. The mgf and CSV files were exported to analyses in GNPS platform [32]. Additionally, metadata were used to describe the ion quantity information in each sample (leaves, stem, and roots from *P. pellucida*).

We used metadata to organize compound information according to the online workflow available on the GNPS documentation [32]. The tolerance of m/z for the precursor ion was adjusted to 0.02 Da and for fragment ion to 0.02 Da. Minimum cosine score above 0.8 and the minimum number of fragment ions fixed on 8. The spectra on the network were then searched in the GNPS spectral libraries. The database spectra were filtered with a minimum cosine score above 0.7 and a minimum of 7 fragment ions correspondence. The results were visualized and organized using Cytoscape version 3.9.2 [33]. To improve the compound classification, we employed MolNetEnhancer [34] and Sirius 4 [35].

## 3. Results

The MolNetEnhancer provided the putative chemical classification of five spectral families that summarize the molecular network of compounds related to extracts from *P. pellucida* leaves, stems, and roots. Therefore, our results allowed us to affirm that we found clusters derived from *C*-glycosyl flavonoids, phenolics compounds, heterocyclics, alkaloids, benzodioxols, and derivatives of fatty amides, as shown in Figure 1B. 

Further annotations were obtained by Sirius 4 [35] considering the systematic classification of the CANOPUS [36]. In addition, taxonomic characteristics reported for *Peperomia* genus were used to enhance confidence of the compound annotation process. Afterward, the dereplication process was carried out, allowing the annotation of twelve compounds (1, 2, 5, 6, 7, 9, 10, 11, 13, 14, 15, and 16), which were described for the first time on this species and, herein, we showed two examples from the benzodioxols group, which are highlighted in Figure 1B. The compounds guineensine (384.2566 *m*/*z*) and pipercallosidine (304.1904 *m*/*z*) were annotated in the GNPS library having already been reported to the Piperaceae family [37]. 

To corroborate with the MS^2^ spectrum mirror plot in the GNPS, Table 1 shows the compound information. For instance, guineensine could be mainly described by product ions of *m*/*z* 311, 283, 175, 161, 135, 123, and 102 [38]. The fragment 311 *m*/*z* likely results from loss of the *N*-Isobutyl group and the 283 by loss of *N*-isobutylformamide. The fragment of *m*/*z* 175 and 161 likely results from heterolytic cleavage in the alkyl chain, followed by the same cleavage at the n_1_ position of the alkyl chain (135 *m*/*z*) [37]. The two last fragments [M + H]^+^ are common to all compounds from the benzodioxols cluster and they characterized the benzodioxol group (123 *m*/*z*) and *N*-isobutylformamide (102 *m*/*z*). Pipercallosidine has already been reported to the Piperaceae family [39] and it was characterized by product ions of *m/z* 231, 213, 187, 173, 159, 135, 123, 111, and 102. Common fragmentations occur for this compound, for instance, a loss of the *N*-Isobutyl group (231 *m*/*z*), cleavage n_1_ (135 *m*/*z*), benzodioxol group (123 *m*/*z*), and *N*-isobutylformamide (102 *m*/*z*).

Through experimental correspondence in the GNPS to guineensine, the compound ([M + H]^+^ of *m/z* 356.2216, C_22_H_29_NO_3_) was named retrofractamide B. The fragments of *m*/*z* 175, 161, 135, 123, and 102 were enough to rank the main candidate. Data from the *Piper nigrum* species corroborated the detection in *P. pellucida* [40]. The ion [M + H]^+^ of *m*/*z* 332.2249 and molecular formula C_20_H_29_NO_3_, already was reported to *Piper nigrum* [41], and herein, it was named as dehydroretrofractamide C (main product ions of *m/z* 173, 161, 135, 123, and 102).

Glycoside fragmentations at ^0,4^X_1_, ^0,3^X_1_, and ^0,2^X_1_ are the main mechanisms (see Figure 2) to characterize the *C*-glycosyl flavonoids. For instance, compounds 2 and 3 presented [M + H]^+^ of *m*/*z* 611.1584 and 565.1524 were annotated as luteolin-6-*C*-glucoside-8-*C*-arabinoside and isoschaftoside, respectively. Isoschaftoside was characterized based on a mirror plot on the GNPS considering all fragments shown in Table 1 and also losses of a portion of hexose (see Figure 2) were confirmed to fragments of *m*/*z* 469 (^0,4^X_1_), 439 (^0,3^X_1_), and 379 (^0,2^X_1_) from the ion of *m*/*z* 499. Compound 3 was previously reported by our group in *P. Pellucida* [42] and the same hexose fragmentations were observed to product ions of *m/z* 455 (^0,4^X_1_), 425 (^0,3^X_1_), and 395 (^0,2^X_1_) likely results from 515 *m*/*z*. The flavonoids cluster can be seen in Figure 3.

To benzene derivatives, peak 8, of *m*/*z* 387.1773 was isolated and reported by us in previous reports [42]. Furthermore, we predict the same structure (see Figure 3) using SIRIUS and it is a dimeric ArC2 [43] named pellucidin B trimethoxy-substituted. Until here there is no fragmentation data in the literature for that structure, but it can likely be mainly explained by product ions of *m*/*z* 219 and 169. A heterolytic cleavage between the aromatic ring and the dihydronaphthalene group explains the product ions of *m*/*z* 169 and 219 *m/z*, respectively. Peak 10 was annotated as 5-methoxy-2-methyl-2,3,8,9-tetrahydrofuro [2,3-h]chromen-4-one and it is described for the first time in *P. pellucida*.

Besides Isoschaftoside and Pellucidin B, other compounds such as Liolide and Pellucidin A have already been reported for this species [42,43]. The others shown in Table 1 such as vidarabine (1) [44], *N*-methylcorydaldine (5), 2′,4′,5′-trihydroxybutyrophenone (6), 2,6-di-*tert*-butyl-4-hydroxymethylphenol (7), velutin (9), and brachystamide B (16) are being reported by our group for the first time for this species after several review process [7,18,44,45,46,47,48,49,50,51,52,53,54].

## 4. Discussion

In this study, we aimed to report the chemical composition of the ethanolic extract from leaves, stem, and roots of *P. pellucida* based on the GNPS library annotation and propagation of these metabolite annotations. In addition, the use of molecular networks (MNs) [32] associated with MolNetEnhancer [34] was helpful in the search for in silico candidates on the Sirius [35] and new compounds to the genus could be described for the first time.

All compounds described here allow us to confirm the chemical diversity for this species as most of these compounds were distributed into five main groups: a large clustering of phenolics and derivatives; a second group with benzodioxols and amides derivatives, ultimately being verified as the common detection of both functional groups in the same metabolite; followed by alkaloids; flavonoids *C*-glycosylated; and lastly, other heterocyclic compounds. Some great fit bioactive examples of these well-annotated compounds include isoschaftoside (3), 2′,4′,5′-trihydroxybutyrophenone (6), velutin (9), pipercallosidine (11), dehydroretrofractamide C (13), retrofractamide B (14), guineensine (15), and brachystamide B (16).

In East Africa, assays using isoschaftoside showed parasitic activity to avoid devastating the maize crop [55]; in another study, the literature highlighted this compound as a potential candidate for the treatment of hypertension [56], and it also showed correlation with antimicrobial activity [57]. 2′,4′,5′-trihydroxybutyrophenone has been applied as food antioxidant [58,59]. Velutin has already been detected in *Piper* species such as *Piper porphyrophyllum* [60], *Piper umbellata* [61], and *Piper clarkii* [62], and it has already been reported as having antibacterial activity [63], anti-melanogenic activity [64], human immunodeficiency virus (HIV-1) activity, and as being against *Trypanosoma cruzi* [65]. In previous reports, pipercallosidine showed anesthetic activity [66], and dehydroretrofractamide C showed inhibition ACAT activity in both rat liver microsomes and HepG2 cancer cells [41]. Retrofractamide B and guineensine promoted gastro-protective effects in experimental animals [67]. In addition, these compounds have shown anti-inflammatory effects in BALB/c mice [37,68], and guineensine demonstrated vasorelaxant activity [69].

Lastly, brachystamide B is already related as being a constituent of the Piperaceae family, namely, the *P. nigrum* and *P. longum* species [38], it is also reported to have anti-inflammatory effects in BALB/c mice [68,70]. Based on previous biological activity information to all compounds described here, the metabolites detected in *P. pellucida* have bioactive potential, some of them have already been confirmed and others are still to be discovered. In this sense, these data make this species a potential new, natural Amazonian source of bioactive compounds.

## 5. Conclusions

The annotation process using machine learning and mirror plot on the GNPS library proved to be efficient, revealing valuable information about the chemical classes, subclasses, and the main bioactive natural compounds of the leaves, stem, and roots from *P. pellucida*. Moreover, it allowed the description of new secondary metabolites in the *Peperomia* genus. Therefore, these data show the potential of *P. pellucida* as a source of discovery of a diversity of bioactive natural compounds and with possible applications in the discovery of new drug candidates.

## Figures and Tables

**Figure 1 molecules-27-01847-f001:**
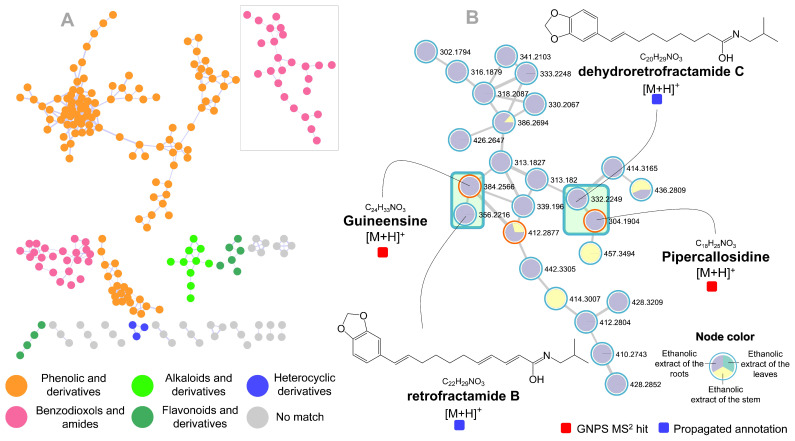
(**A**) Classification of compounds classes and subclasses of the ethanolic extract from *P. pellucida* leaves, stem, and roots using MolNetEnhancer [34]; (**B**) Some examples of propagated annotation using Sirius 4 [35].

**Figure 2 molecules-27-01847-f002:**
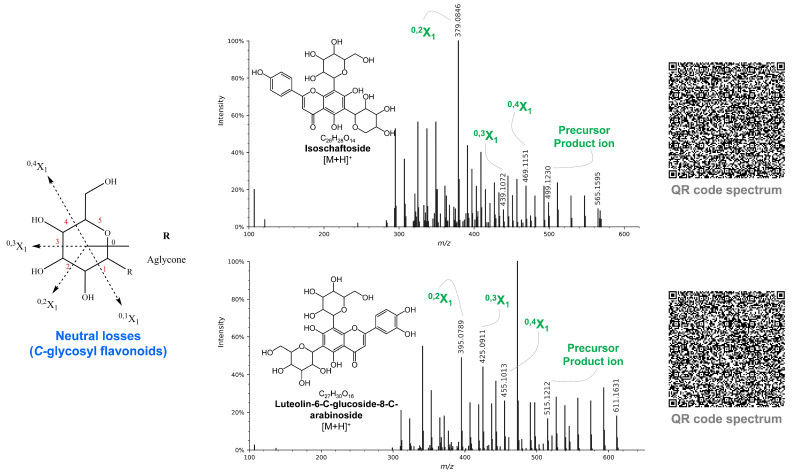
Characteristic losses of portions of hexose in flavonoids (e.g., MS/MS spectrum to Isoschaftoside and Luteolin-6-*C*-glucoside-8-*C*-arabinoside). Both mass spectra are available by QR code.

**Figure 3 molecules-27-01847-f003:**
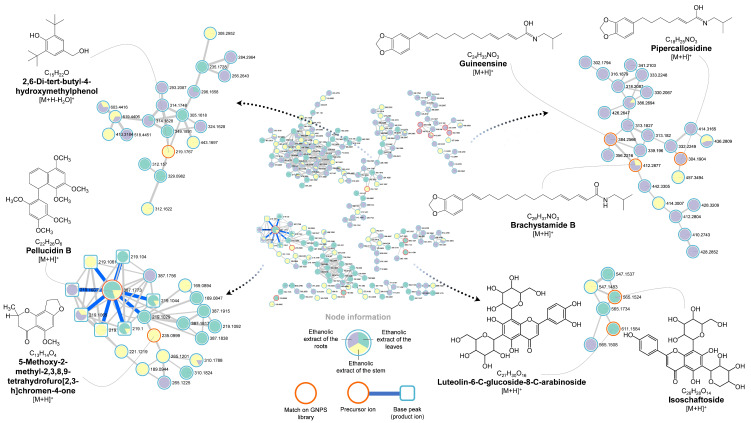
Selected clusters and annotation of compounds from the molecular network in positive ion mode for the ethanolic extract of the leaves, stem, and roots from *P. pellucida*. Numbers beside the nodes correspond to the accurate precursor mass (*m*/*z*), red circle: MS2 match on GNPS library, node color blue correspondent to leaves, yellow correspondent to stem, and blue correspondent to roots.

**Table 1 molecules-27-01847-t001:** UHPLC-MS/MS data of the main compounds annotated and propagated for each group.

Peak	Rt (min)	Molecular	[M + H]^+^ (*m*/*z*)			Main Product Ions (MS/MS)	Putative Compound
		Formula	Calculated	Accurate	mDa		
**1**	0.55	C_10_H_13_N_5_O_4_	268.1046	268.102	2.6	**136**, 119	vidarabine
**2**	2	C_27_H_30_O_16_	611.1612	611.1584	2.8	593, 575, 557, 539, 527, 515, 491, **473**, 455, 443, 425, 395, 371, 341, 311	luteolin-6-*C*-glucoside-8-*C*-arabinoside
**3**	2.34	C_26_H_28_O_14_	565.1557	565.1524	3.3	547, 529, **511**, 499, 481, 469, 457, 439 427, 409, 397, 379, 349, 337, 325, 307, 295	isoschaftoside
**4**	3.41	C_11_H_16_O_3_	197.1178	197.1189	1.1	179, 161, **133**	liolide
**5**	3.78	C_12_H_15_NO_3_	222.113	222.1116	1.4	204, **165**, 150, 133, 105	*N*-methylcorydaldine
**6**	4.04	C_10_H_12_O_4_	197.0814	197.083	1.6	182, 169, 154, **138**, 123, 111	2′,4′,5′-trihydroxybutyrophenone
**7**	5.26	C_15_H_22_O	219.1749	219.1767	1.8	203, 187, 173, 161, 145, 133, **119**, 105	2,6-di-*tert*-butyl-4-hydroxymethylphenol
**8**	6.18	C_22_H_26_O_6_	387.1808	387.1773	3.5	**219**, 169, 145	pellucidin B
**9**	6.27	C_17_H_14_O_6_	315.0869	315.0869	0	**300**, 272, 257, 243, 201, 187, 167, 149	velutin
**10**	7	C_13_H_14_O_4_	235.097	235.0999	2.9	205, **189**, 177, 161, 146, 133, 118, 105	5-methoxy-2-methyl-2,3,8,9-tetrahydrofuro [2,3-h]chromen-4-one
**11**	7.18	C_18_H_25_NO_3_	304.1913	304.1904	0.9	231, 213, 187, 173, 159, **135**, 123, 111, 102	pipercallosidine
**12**	7.77	C_22_H_28_O_6_	389.1964	389.2002	3.8	315, **221**, 190, 181, 174, 169, 147, 129, 114, 105	pellucidin A
**13**	8.2	C_20_H_29_NO_3_	332.2226	332.2249	2.3	259, 161, 149, 135, 123, 102	dehydroretrofractamide C
**14**	8.52	C_22_H_29_NO_3_	356.2226	356.2216	1	283, 269, 215, 187, 175, 167, 161, 149, **135**, 123, 102	retrofractamide B
**15**	9.39	C_24_H_33_NO_3_	384.2539	384.2566	2.7	311, 283, 175, 161, **135**, 123, 102	guineensine
**16**	10.3	C_26_H_37_NO_3_	412.2852	412.2877	2.5	339, 311, 290, 203, 185, 175, 161, 149, **135**, 123, 102	brachystamide B

Note: mDa, mass defect (millidalton); Bold product ion means the base peak from MS/MS spectrum.

## Data Availability

All support data used in this study are available from the authors.

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
