# Peer review of "Chemical Composition of Leaves, Stem, and Roots of Peperomia pellucida (L.) Kunth"

_molecules, 2022, doi:10.3390/molecules27061847_

Round 1

Reviewer 1 Report

Te aim of the present study was focused on assessment of the chemical  composition of Peperomia pellucida (L.) ethanolic extract. I have a few remarks about this study. The first aspect refers to the abstract. In the abstract there is an affirmation (lines 22-26) - several bioactive compounds were detected for the first time in the Peperomia genus, as such antioxidant (2',4',5'-Trihydroxybutyrophenone), antibacterial (velutin), activity against HepG2 cancer cells (dehydroretrofractamide C), and gastro-protective effect (retrofractamide B) - which is not actually demonstrated in this article. There are also certain aspects that are missing in the description of the methods. For example, the concentration of ethanol that was used for extraction is missing. It is indicated a method from the literature after which the experiment was performed, it is a review and I am not sure which method was selected for the design of this study. The title of the article refers to the identification of biactive components; The chemical composition of the alcoholic extracts of P. pellucida prepared from leaves, stem, and roots has been described but without evaluating the biological effect of these components. The authors state that the biological effects of some components have already been confirmed, and others have yet to be discovered. Yes, this species can become a source of bioactive compounds, but research needs to be done.

Author Response

Reviewer: Te aim of the present study was focused on assessment of the chemical composition of Peperomia pellucida (L.) ethanolic extract. I have a few remarks about this study. The first aspect refers to the abstract. In the abstract there is an affirmation (lines 22-26) - several bioactive compounds were detected for the first time in the Peperomia genus, as such antioxidant (2',4',5'-Trihydroxybutyrophenone), antibacterial (velutin), activity against HepG2 cancer cells (dehydroretrofractamide C), and gastro-protective effect (retrofractamide B) - which is not actually demonstrated in this article. 

The title of the article refers to the identification of bioactive components; The chemical composition of the alcoholic extracts of P. pellucida prepared from leaves, stem, and roots has been described but without evaluating the biological effect of these components.

The authors state that the biological effects of some components have already been confirmed, and others have yet to be discovered. Yes, this species can become a source of bioactive compounds, but research needs to be done.

There are also certain aspects that are missing in the description of the methods. For example, the concentration of ethanol that was used for extraction is missing.

It is indicated a method from the literature after which the experiment was performed, it is a review and I am not sure which method was selected for the design of this study.

Authors – Dear reviewer, we greatly appreciate your contributions to improve the presentation of our work. After your comments, we realized how pretentious it seemed on our part to use terms referring to the activity we did not perform. Such terms were, therefore, removed from abstract and the activities of the compounds mentioned by us were informed in the discussion of the results with their respective citations. For the same reason we drastically modified the title too. We hope that this time the writing is representative of our work. Besides, some details from methods you have asked were described. Thank you for your valuable contributions.

Reviewer 2 Report

The manuscript “Natural bioactive compounds from leaves, stem, and roots of Peperomia pellucida (L.) Kunth” was submitted to Molecules for publication.

Broad comments:

The study describes the identification of several compounds from an Amanzonian medicinal plant by molecular networking and machine learning approaches. The experiments are well described and the conclusions reported from the molecular networks sound reasonable. The only major point from my side that must be revised is the language. I strongly suggest to engage a native speaker and therefore increase the readability of the paper. One such example is the first sentence of the introduction. Here it is not clear what is meant by “the second most biology and chemical diversity genus”.

Specific comments:

Line 43:                              Please write “chemodiversity” instead of “chemiodiversity”.

Line 45:                              Is it “erva-de-jaboti” or “erva-de-jabuti” as written in the abstract?

Table 1, Peak 7:                Please write “tert” in italics

Table 1, Peak 10:             Please write “5-methoxy-…” in small letters

Line 221:                           Please write “2’,4’,5’-trihydroxybutyrophenone” and “velutin” in small letters and keep this throughout the manuscript.

Line 224:                           Please write “In East Africa”

Line 231:                           Please write “Pipercallosidine” in big letters as it is the beginning of the sentence.

General:                            Keep a space between before the references.

Author Response

Reviewer: Comments and Suggestions for Authors

The manuscript “Natural bioactive compounds from leaves, stem, and roots of Peperomia pellucida (L.) Kunth” was submitted to Molecules for publication.

Broad comments:

The study describes the identification of several compounds from an Amanzonian medicinal plant by molecular networking and machine learning approaches. The experiments are well described and the conclusions reported from the molecular networks sound reasonable. The only major point from my side that must be revised is the language. I strongly suggest to engage a native speaker and therefore increase the readability of the paper. One such example is the first sentence of the introduction. Here it is not clear what is meant by “the second most biology and chemical diversity genus”.

Authors – Dear reviewer, thank you very much for your valuable contribution to the improvement of our manuscript. We carefully evaluate your recommendations and made the indicated corrections, which we highlighted in change control in the new version submitted for your conference. We also took the opportunity to re-write and check other points that are included in the new version submitted.

Specific comments

Reviewer: Line 43: Please write “chemodiversity” instead of “chemiodiversity”.

Authors – Thank you for your comment. This word was corrected to chemiodiversity.

Reviewer: Line 45: Is it “erva-de-jaboti” or “erva-de-jabuti” as written in the abstract?

Authors – The popular name “erva-de-jabuti” is written correctly.

Reviewer: Table 1, Peak 7: Please write “tert” in italics

Authors – This was corrected in the whole manuscript.

Reviewer: Table 1, Peak 10: Please write “5-methoxy-…” in small letters

Authors – This was corrected in the whole manuscript.

Reviewer: Line 221: Please write “2’,4’,5’-trihydroxybutyrophenone” and “velutin” in small letters and keep this throughout the manuscript.

Authors – Thank you. These words were corrected in the whole manuscript.

Reviewer: Line 224: Please write “In East Africa”

Authors – Thank you. Now it is written as “In East Africa”.

Reviewer: Line 231: Please write “Pipercallosidine” in big letters as it is the beginning of the sentence.

Authors – We appreciate your correction. Done!

Reviewer: General: Keep a space between before the references.

Authors – The space was checked and corrected to rules of molecules

Round 2

Reviewer 1 Report

Yes, minor improvements have been made. The title has been modified and certain aspects related to the extraction protocols.As I said, I don't feel qualified to judge about the English language and style, but it would be necessary because there are mistakes in expression.